# The challenge of involving old patients with polypharmacy in their medication during hospitalization in a medical emergency department: An ethnographic study

Pia Keinicke Fabricius[1,2]*, Ove Andersen[1,2,3], Karina Dahl Steffensen[4,5,6], Jeanette Wassar Kirk[1,7]

1 Department of Clinical Research, Copenhagen University Hospital, Amager and Hvidovre, Copenhagen, Denmark, 2 Department of Clinical Medicine, University of Copenhagen, Copenhagen, Denmark, 3 Emergency Department, Copenhagen University Hospital Hvidovre, Hvidovre, Denmark, 4 Department of Oncology, Lillebaelt Hospital, University Hospital of Southern Denmark, Vejle, Denmark, 5 Institute of Regional Health Research, University of Southern Denmark, Odense, Denmark, 6 Center for Shared Decision Making, Lillebaelt Hospital, University Hospital of Southern Denmark, Vejle, Denmark, 7 Department of Public Health, Nursing, Aarhus University, Aarhus C, Denmark

* pia.keinicke.fabricius@regionh.dk

**Data Availability Statement:** The data that support the findings of this study contain potentially identifying or sensitive information about

## Abstract

### Background

More than 70% of patients admitted to emergency departments (EDs) in Denmark are older patients with multimorbidity and polypharmacy vulnerable to adverse events and poor outcomes. Research suggests that patient involvement and shared decision-making (SDM) could optimize the treatment of older patients with polypharmacy. The patients become more aware of potential outcomes and, therefore, often tend to choose less medication. However, implementing SDM in clinical practice is challenging if it does not fit into existing workflows and healthcare systems.

### Aim

The aim was to explore the determinants of patient involvement in decisions made in the ED about the patient's medication.

### Methods

The design was a qualitative ethnographic study. We observed forty-eight multidisciplinary healthcare professionals in two medical EDs focusing on medication processes and patient involvement in medication. Based on field notes, we developed a semi-structured interview guide. We conducted 20 semi-structured interviews with healthcare professionals to elaborate on the findings. Data were analyzed with thematic analyses.

### Findings

We found five themes (determinants) which affected patient involvement in decisions about medicine in the ED: 1) blurred roles among multidisciplinary healthcare professionals, 2)

healthcare professionals, managers and their departments. The data are not publicly available due to ethical restrictions. The research Data Protection Agency of the Capital Region of Denmark can be contacted for further requests: E-mail: videnscenterfordataanmeldelser. rigshospitalet@regionh.dk or Phone: +45 35 45 52 11.

**Funding:** Ove Andersen received funding provided by Velux Foundation grant number (00021736): https://veluxfoundations.dk/da/forskning/ aldringsforskning. The funders had no role in the study design, data collection and analysis, decision to publish, or preparation of the manuscript.

**Competing interests:** The authors have declared that no competing interests exist.

older patients with polypharmacy increase complexity, 3) time pressure, 4) faulty IT- systems, and 5) the medicine list as a missed enabler of patient involvement.

## Conclusion

There are several barriers to patient involvement in decisions about medicine in the ED and some facilitators. A tailored medication conversation guide based on the SDM methodology combined with the patient's printed medicine list and well-functioning IT- systems can function as a boundary object, ensuring the treatment is optimized and aligned with the patient's preferences and goals.

## Introduction

Globally the prevalence of patients with polypharmacy is set to rise as the population ages and more people suffer from multiple long-term conditions taking numerous medicines. Consequently, it makes more people prone to drug-related events and frequently acute admissions in emergency departments [1, 2]. More than 1,000,000 out of 1,300,000 yearly hospital admissions in Denmark are acute admissions in emergency departments (EDs), and more than 70% of these admissions are older patients (65 + years) with comorbidity conditions and polypharmacy [3, 4], defined as taking five or more medications [5]. We have earlier identified that the use of potentially inappropriate use of medicine (so-called PIMS) in older patients (65 + years) admitted to our ED was very common (a prevalence of 85%). Furthermore, the use of PIMS was associated with low functional status and reduced health-related quality of life [6].

In recent years, there has been an increasing focus on dealing with patients with polypharmacy, as treatment is getting more and more complex [1]. This complexity is also seen inside hospitals, where historically, the medication process has been a simple task performed mainly by physicians and nurses [7], but in recent years, more healthcare professionals with different roles and responsibilities for medication have been introduced into hospitals; e.g., pharmacists have been introduced into ED departments to optimize rational use of medication, and ensure patient safety [8]. Polypharmacy represents a global patient safety risk, but according to the World Health Organization (WHO), patients can play a key role in the early detection of inappropriate polypharmacy if they are involved and invited to tell more about their symptoms and side-effects [1]. Inappropriate polypharmacy is defined as when medicines are no longer needed, cause adverse drug reactions, or the patient is not willing or able to take medicines as intended [1].

Another problem with polypharmacy is that most clinical guidelines typically have a strong focus on starting medication with limited guidance about not starting, reducing, or stopping medication (known as deprescribing) [9]. Furthermore, most clinical guidelines do not offer specific guidance to how "decisions" about medicine should be made with polymedicated patients, reflecting the patient's preferences, needs, and values to support patient involvement optimally and shared decision making [10]. The term "Patient involvement" is often used interchangeably and covers a diverse range of possibilities, emphases, models, and practices, with Shared decision Making (SDM) being one of the leading concepts within [10]. SDM is cited to be the pinnacle of patient-centered care [11] and is a specific approach where healthcare professionals and patients build a partnership, where the healthcare professionals share the best available evidence, and the patients express their values and preferences and participate in decisions about their medical treatment [9]. Research shows that involving patients in their care and listening to their views improves knowledge, decision outcomes, compliance

with treatments [12]. However, there is still a lack of solid studies investigating the impact of patient involvement of older multimorbidity patients with polypharmacy.

Ideally, SDM with older polymedicated patients in the ED would be a process where the older patients are "active" partners. The patient works in partnership with the healthcare professionals about treatment choices. The healthcare professional presents different options using risk communication, and the patient's preferences are explored and supported, and decisions about changes (or no changes) to the patient's medicine are reached jointly [10].

Patient-centered healthcare and patient involvement is a quality goal in many countries' healthcare policies, including Denmark's [10, 13]. Nevertheless, the gap between healthcare policy and routine clinical practice is persistent. Studies indicate that although there have been some promising signs of improvement, patients do not always feel as involved in their treatment as they would like to be, and patients do not always receive as much information about their medicine as they would like to have [10].

The healthcare professionals' attitude towards patient involvement is also cited as a barrier while some healthcare professionals may have the assumptions that the older patients cannot contribute to decisions about their medicine, or the healthcare professionals think that they know their patient's preferences already, so they do not need to ask the patient [14]. Additionally, older patients may have different preferences for involvement, low health literacy, difficulty hearing, or cognitive impairment, which makes it more challenging to become involved in decisions about medicines [15].

Furthermore, several studies highlight the influence of contextual factors influencing the implementation of patient involvement and SDM in routine practice, which could fail if it is not implemented in a way that fits into existing workflows, organizations, and health systems [16]. Although the general principles of SDM have been used partly by emergency physicians ad hoc for decades, the systematic use and evaluation of SDM in the Emergency Department (ED) still remains in its infancy [14]. Therefore the aim of this study was to explore the determinants of patient involvement in decisions made in the ED about the patient's medication.

## Materials and methods

This study was the first in a program called OPTICARE. OPTICARE's aim is to develop a "Medication Conversation Guide" that uses an SDM methodology tailored to involve polymedicated 75+-year-old patients in the ED in their medication (age 75 + year was put in place due to the wording in the Velux Foundation, the fund achieved).

This study was a baseline study with the aim of exploring the determinants of patient involvement in decisions about their medication before implementing SDM. Determinants are defined as "Factors of practice which either prevent or enable improvements in professional healthcare practice" [17]. Because implementation studies recognize the importance of culture and organizational context [18], we conducted an ethnographic study with a focus on the healthcare professionals who treat patients in the specific context of the ED. We did not expect to observe SDM in the ED as SDM has not yet, been implemented in our medical EDs. We, therefore, focused on patient involvement, which is more broadly defined by Cribb [10] as "active rather than passive patients." When exploring patient involvement in decisions about medicine in the ED, we focused on the patient's active role in decisions about their medicine in the medical ED and determinants influencing this active involvement.

### Study setting

The study was conducted at a university hospital in the capital region of Denmark, where the healthcare system is publicly funded by taxes. The Danish welfare state provides free treatment

for all citizens requiring medical care, as well as free hospital and home-based care services. The acute medical patients are referred to the hospital either by the general practitioners (GPs), emergency medical helplines, after-hours GP services, or ambulances.

The university hospital is divided into three different locations which each houses a medical emergency department to cover the catchment area of 517,00 people. The context in this study includes two of the medical emergency departments, which are named (Department X and Department Y) (see Table 1).

The EDs have an average daily intake of approximately 30–45 patients with hospitalization of up to 48 hours in the ED.

For over a decade, emergency services across Denmark have changed due to health policy reforms with the implementation of new EDs (so-called Acute Medical Units) with the aim of improving the quality and efficiency of emergency care [19]. The new EDs offer a single point of hospital entry for all emergency care (except children and women in labor), 24/7 for effective emergency diagnostics and treatment with the continuous presence of skilled senior physicians, which covers ED physicians and senior physicians from the medical specialties.

Several other countries as the UK, Australia, the Netherlands, Germany, and France, have introduced similar reforms to secure safe and efficient pathways for patients in need of emergency care [20].

The new Danish EDs treat patients for up to 48 hours before discharge to home or a specialty department. It is an overall goal that 70% of all acute patients are discharged directly from the new ED without further hospitalization to a medical ward. This also includes the acute admitted older patients with multimorbidity and polypharmacy.

As the new ED is expected to be implemented in 2023, this baseline study serves as exploring new ways to improve the future patient pathways in the new EDs for older patients with polypharmacy.

Since 2014, Department X has had clinical pharmacy technicians (in Denmark called pharmaconomists) employed who are responsible for the administration of medication during dayshift. The pharmaconomists hold a three-year degree and are comparable to most pharmacy technicians in other countries [21]. In Denmark, pharmaconomists work at their own private pharmacies, and in hospital pharmacies, where they often are responsible for the daily clinical management of

**Table 1. Setting characteristics.**

| |
|---|
| **Setting: Department X** |
| • Department X is one of the largest EDs in Denmark and is located in a hospital that is classified as an acute hospital receiving critically ill patients into its Intensive Care Unit (ICU). |
| • The department has 29 beds. |
| • There are approximately 4–5 ED senior physicians. |
| • There are approximately 8–11 physicians are employed in one of the hospital's medical specialty wards covering shifts in the ED. |
| • There is a geriatric team with geriatric doctors and nurses. |
| • Pharmacists and pharmaconomists administer medication during the dayshift on weekdays. |
| **Setting: Department Y** |
| • Department Y is a smaller ED than X, and its location is not classified as an acute hospital and does not have an ICU. |
| • The department has 19 beds. |
| • There are 2 ED senior physicians. |
| • 9 senior physicians are employed in one of the hospital's medical specialty wards and cover shifts in the ED, where they treat all hospitalized patients regardless of their medical specialty. |
| • Nurses manage medication with minor involvement from a pharmacist. |

medicine in the wards. Their work includes administering medicine to the hospitalized patients in the ward. Like the pharmacy technicians in the UK, the pharmaconomists in Denmark practice dispensing medicine without the supervision of a pharmacist. In September 2018, pharmacists were introduced into the ED with the aim of supporting ED physicians with the implementation of an e-health platform, which is used while the patient is hospitalized. When the patient is not hospitalized, the prescribed medicine is registered in the patient's Shared Medication Card (SMC), which is housed in a central database that contains electronic data on all Danish citizens' prescriptions and medicine [22]. The Pharmacists and the geriatric physician do structured medication reviews on a daily basis for selected older polymedicated patients in the ED.

## Study design

The design was a qualitative ethnographic study [23] carried out in three steps. First, we conducted field observations. Based on the observations, we developed a semi-structured interview guide attached in the S1 File. Then, we conducted semi-structured interviews with healthcare professionals [24] from both EDs. The study was structured according to the standards for reporting qualitative research (SRQR) [25].

## Participants

Nonprobability purposeful sampling with maximum variation [26] resulted in the recruitment of 48 different healthcare professionals (see Table 2) representing a broadly mixed multidisciplinary team who were observed for 58 days.

## Data collection and analysis

### Field observations

Initially, a field study was carried out in Department X from October 2018 to February 2019, in which the researcher (P.F.) acted as a nonparticipant observer [23] for 33 days. Influenced

**Table 2. Participant characteristics.**

| Data | Department X (n) | Department Y (n) | Total (n) |
|---|---|---|---|
| Field observations | | | |
| ED physicians | 8 | 15 | 23 |
| Nurses | 7 | 7 | 14 |
| Pharmacists | 5 | - | 5 |
| Pharmaconomists | 5 | - | 5 |
| Geriatric physicians | 5 | - | 5 |
| Secretaries | 2 | 1 | 3 |
| Geriatric nurses | 2 | - | 2 |
| Consulting physicians from the hospital's medical specialties | 1 | - | 1 |
| **Total** | **35** | **23** | **58** |
| Semi structured interviews | | | |
| ED physicians | 4 | 4 | 8 |
| Nurses | 4 | 2 | 6 |
| Pharmacists | 1 | - | 1 |
| Pharmaconomists | 2 | - | 2 |
| Geriatric physicians | 1 | - | 1 |
| Secretaries | 1 | - | 1 |
| Geriatric nurses | 1 | - | 1 |
| **Total** | **14** | **6** | **20** |

by J. Spradley's participant observation approach [23], an observation guide was developed as a broad, descriptive guide to observation to locate social situations in which medication practices occurred. The focus points were as follows: What is going on, and who is participating in the medication process? What do people do and say about medication? Who makes decisions regarding medication? What or who is involved in those decisions? What do the patients say about medication? What role do patients or their relatives have? The observation approach was in accordance with Spradley's description of the "mini-tour," as from the start, we had chosen a narrow focus on medication practices [25] which during the field observations became even more focused and selective. We chose Spradley's participant observation approach while Spradley is a classic within ethnographic fieldwork. Furthermore, Spradley offers a structured stepwise approach to participant observations, which is helpful for ethnographers. Identical field studies were conducted by the same researcher in Department Y from August 2019 to September 2019 for 23 days. In total, 48 different healthcare professionals were followed in their daily work rounds for three hours on average. Researcher P.F. observed the healthcare professionals for 58 days in 144 hours.

The healthcare professionals were followed during day and evening shifts. The recruitment of participants was planned in advance with the managing nurses, who helped to locate participants according to our selection criteria. These criteria included participants from different healthcare professions involved in the patient's medicine and with the most diverse years of clinical competencies to get as nuanced insight as possible of factors influencing patient involvement in decisions about their medicine. From the start, we had the assumption that all healthcare professionals in the ED (except the physiotherapists and secretaries) were involved in the medication process and therefore were relevant informants who were included based on their profession (ED physicians, geriatric physicians, consulting physicians from the medical specialties, nurses, pharmacists, and pharmaconomists).

The observational approach was nonparticipatory [23]. Sometimes asking clarifying questions in go-along interviews allow the healthcare professional to clarify what was going on and to express reflexive aspects of their experience in situ. Detailed field notes were taken, documenting, as close as possible, what was said and done by whom. Verbal and nonverbal reactions, e.g., bodily reactions, facial expressions, moods, and sounds, were documented on a paper block with a pen. Immediately after the observation, the written field notes were further expanded in a word document. Data from the field observations consist of 221 A4 pages of rich text material.

To capture the researcher's thoughts and how her readings of other SDM studies affected the ongoing investigation, the researcher's thoughts were documented in a research diary [25] which was frequently discussed with co-author J.K.

## Interview guide

We developed a semi-structured interview guide [24] from an initial analysis of the field notes. The overall aim of the interview guide was to let the healthcare professionals elaborate on their perspective on the determinants of patient involvement in medication that had been identified through the field observations. The interview guide was designed in collaboration with J.K.

Before the semi-structured interviews were carried out, the interview guide was pilot tested for comprehensibility with two non-clinical colleagues. The pilot test resulted in minor changes, e.g., how questions were worded before the interviews were carried out.

## Semi-structured interviews

To strengthen the plausibility and credibility of the findings, which is a validity criterion in ethnographic research [25], semi-structured interviews were also conducted to test the researcher's interpretations of the field observations.

Twenty ED healthcare professionals were interviewed by the first author in January 2020. The selection of respondents was based on two inclusion criteria. Ten healthcare professionals who were followed during the field study and ten healthcare professionals who were not followed were selected in order to achieve as much nuance in the perspectives as possible. The interviews took place in a room closely connected to the two EDs and lasted from 25 to 45 minutes. Each interview was tape-recorded and later transcribed verbatim by P.F. (n = 7) and three research secretaries (n = 13). In total, the data material consisted of 183 A4 pages.

## Data analysis

A thematic analysis was carried out, which is a qualitative analytic method that offers reflexivity unconstrained by a pre-existing theoretical frame [27].

The analysis was carried out in two steps. The first author, P.F., read and reread the field notes to become familiar with the material [25], The field notes were initially analyzed with an open, inductive, thematic approach [27]. Text pieces were coded and recoded in an ongoing iterative interpretation process [29] using the software program NVIVO 11 [28]. The coding process was frequently read and discussed with J.K. and later discussed with the rest of the research team to ensure the credibility of the analysis [29]. The initial analysis of the field notes resulted in the themes that informed the questions in the semi-structured interview guide. The themes' primary focus was on the healthcare professionals' perspectives on barriers and facilitators (the determinants) of patient involvement in decisions about medicine in the ED.

In the second step, the transcribed text from the interviews was initially coded and merged deductively into the existing themes from the interview guide. See Table 3. Data saturation was reached after 20 interviews, as no further issues were identified in the data material [29]. During an iterative interpretation process with J.K., some themes were merged while others were divided into new themes [29] as it became apparent that some themes appeared to be more important for patient involvement in decisions about medication than others.

During the analysis and interpretation process, it became clear to us that medication (and especially polypharmacy) crosses boundaries between different healthcare sectors, different departments, medical specialists, and medication has similarities with the sociological theory of boundary objects. Boundary objects has the ability to transmit knowledge and meaning between different group of people [30, 31] and our results will be discussed through this theoretical lens in the discussion section.

**Table 3. Analysis example from interview transcription to theme.**

| Interview Transskription | Code | Sub-theme | Theme |
|---|---|---|---|
| *". . . but it's obvious that there are two IT- systems that don't work well together, and that they do not work well together does not make the situation concerning the patient's medication any easier . . . if just . . . if just all IT worked well, and it was possible to figure out what patients actually took at home. Often different medication is registered in the systems, and the larger the background issues, such as IT and such, then focus is on those issues instead of the patient."* (Interview, Pharmacist, Department X) | Poor integration between the e-health platform and SMC took time and attention away from direct contact between the physician and patient. | • Poorly integrated IT systems lead to less patient involvement | Faulty IT- systems |

## Ethical issues

The Danish Data Protection Agency (file number VD-2019-264) approved this study. Under Danish law, no further formal approval from the Ethics Committee is necessary for studies that do not involve biomedical issues. Written consent was obtained from the departmental management of both EDs. Written and oral informed consent was also collected from the healthcare professionals who were observed and interviewed. All participants were guaranteed anonymity and the opportunity to withdraw their consent to the study at any time. With this, our study complies with the World Medical Associations´ ethical principles for medical research involving human subjects (The Declaration of Helsinki) [32]. In the field observations, P.F. acted in line with situational ethics, balancing intuition, sense, morality, and responsibility [33], constantly judging if the researchers' presence could violate the patient's integrity. Occasionally, P.F. judged a patient to be too vulnerable and in too severe a condition that it would have been unethical to continue the observation. The researcher, P.F., has worked as a nurse and consultant at the hospital for 20 years, which led to a position in which it was relatively easy to obtain access and insights into the healthcare professionals' thoughts on medication and patient involvement.

## Results

The observations and interviews revealed that managing medication to- and communication with the patients in the medical ED is a fragmented process involving various healthcare professionals, medical specialties, time pressure, and faulty IT- systems, which influence patient involvement in medication decisions and may impede medication optimization. Further, the study revealed that the patients' printed medicine list could facilitate more communication about medicine and increase patient involvement in decisions about medicine in the ED if the healthcare professionals were more aware of this opportunity.

We identified five themes (determinants) which affected patient involvement in decisions about medicine in the ED:

1. Blurred roles among multidisciplinary healthcare professionals,

2. Older patients with polypharmacy increase complexity,

3. Time pressure,

4. Faulty IT- systems, and

5. The medicine list as a missed enabler of patient involvement.

Four of the themes identified issues that prevented patient involvement in decisions about medication, while one of the themes identified a solution that could enable patient involvement if healthcare professionals were aware of this opportunity. The five themes include eight sub-themes illustrating how the themes influence patient involvement in decisions about medicine in the ED (See Table 4).

### Theme 1: Blurred roles among multidisciplinary healthcare professionals

The observations and interviews revealed that managing medication in the ED is a fragmented process that involves different healthcare professionals: physicians, pharmacists, pharmaconomists, and nurses with different competencies, which influenced patient involvement.

Unlike the physicians and nurses, the pharmacists and pharmaconomists were only focused on medication, and according to the pharmacists and pharmaconomists themselves, they therefore had the resources to help the physicians, who often had difficulty obtaining an

Table 4. Themes and subthemes.

| No. | Themes | Sub-themes |
|---|---|---|
| 1. | Blurred roles among multidisciplinary healthcare professionals | • Invisible roles |
| | | • Responsibility for polypharmacy |
| 2. | Older patients with polypharmacy increased complexity | • Trust |
| 3. | Time pressure | • A revolving door |
| | | • Time-consuming communication |
| 4. | Faulty IT-systems | • Poorly integrated IT systems lead to less patient involvement |
| | | • Challenging discharge |
| 5. | The medicine list as a missed enabler of patient involvement | • Inconsistently use of medicine lists |

overview of patient medication from the e-health platform and the Shared Medication Card (SMC). In addition, the observations revealed that the pharmacists and the pharmaconomists had found several faulty medication prescriptions that were corrected before medication was administered.

One pharmaconomist stated:

*"We only focus on the medicine. We do not have to find out when they [the patient] will be discharged or if they should go to a nursing home or some outpatient clinic."* (Interview, Pharmaconomist, Department X).

**Sub-theme: Invisible roles.** In contrast, some professionals were not aware of their role in the medication process, even though they often had significant roles in decisions about the patient's medicine. Several nurses and some medical secretaries stated that they had little to do with the patients' medication. However, the observations showed the opposite. The nurses had frequent conversations with the patients about medicine, and often it was the nurses who knew why a patient did not take their medicine as intended. The nurses passed on the information to the physicians, who made the medication changes based on the information from the nurses. Therefore, nurses often had a decisively significant influence in decisions about their patient's medication in the ED, even though many nurses were unaware of it.

Similarly, the observations revealed that the secretaries often had the time-consuming work of following up on loose ends related to a medication when patients called the ward after discharge.

One of the secretaries said: *"We are like intermediaries spending a lot of time on things we have not been involved in earlier."* (Observation, Secretary, Department X).

The interview revealed that the secretaries often had the task of follow-up communication about medicine with patients calling the ED after discharge and were unaware of why their medication had been changed. It happened quite often, according to the secretaries.

**Sub-theme: Responsibility for polypharmacy.** The physicians often preferred not to evaluate a polymedicated patient's usual medications as they did not always perceive that it was their job but the patient's GPs job. In these situations, very limited patient communication about medicine was observed. Only when it was suspected that the medicine caused the patient's acute situation, the ED physicians had many questions to the patient about the medicine. However, some ED physicians felt bad about not evaluating the patient's polypharmacy, and they were aware that some GPs expected the same as the physician responsible for treatment in the acute care setting. One chief physician said:

*"Unfortunately, that's the culture. Everyone thinks that somebody else should do it."* (Observation, Chief physician, Department Y).

During the observations, some pharmacists and pharmaconiomists raised the concern that polymedicated patients are more frequently acute hospitalized than patients without polypharmacy. The pharmacists emphasized the possibility of a structured medication review with patient involvement and SDM in the ED, for all, and not only a few older polymedicated patients.

Nevertheless, the observations and the interviews revealed that the pharmacists, pharmaconomists, nurses, physicians, and secretaries all had important roles in ensuring patient safety and patient involvement in decisions about medication, even though there were different perspectives on their efforts and responsibilities for patient involvement in decisions about medicine in the ED.

## Theme 2: Older patients with polypharmacy increased complexity

The observations and interviews revealed that the complexity, fragmentation, and communication were exacerbated in older patients with polypharmacy. In addition, the observations and the interviews revealed that older patients' polypharmacy is challenging in the ED because the' medication registrations in the SMC could not be trusted, which complicated patient involvement.

**Sub-theme: Trust.** Physicians had to take the patients' SMC medication records at face value, and most assumed that the patients took the medications as registered. However, the observations showed that very few older patients knew what was registered on their SMC, and often the SMC had not been updated by the patient's GP for a long time. This resulted in ambiguities and complications in communication about medication, which was also time-consuming.

The ED physicians found themselves in a dilemma regarding whether they should rely on the SMC or the older patients' own account of their medication, and the observations showed that there could be errors either way, as the older patients sometimes had a hard time remembering their actual medicine. In the following situation, an older patient informed the physician that she was not given any medication, and when the physician subsequently read about five fixed drugs in the SMC, the physician spontaneously said to P.F.:

*"Look, and she said she did not get any medication!"* (Observation, Physician, Department Y).

In the interview, a younger physician explained that he deliberately did not involve older patients with polypharmacy in decisions about their medicine because he had experienced that older patients often did not know about their medicine or did not care about it, and therefore he did not try.

The observation revealed that the ED physicians were often reluctant to trust the older patients' accounts of their own medications, as the older patients were in an acute condition when admitted to the ED. This acute situation affected how much dialogue and patient involvement about medication was observed because many physicians were reluctant to involve the older patients. One physician, before seeing a patient with polypharmacy for the first time, exclaimed:

*"What can you expect from an 87-year-old?"* (Observation, Physician, Department Y).

However, some physicians made a great effort to involve the older patients with polypharmacy at the physicians round. Sometimes it seemed to be a success, and there was dialogue,

and the patient had an influence on decisions about which medicine should be deprescribed before being discharged. The observations revealed that older polymedicated patients who were active asking questions about their medicine seemed to have high health literacy were easier to involve because they often involved themselves. This highlights that older patients have different preconditions for involvement in decisions about their own medicine. To secure a good relationship between the patient and the healthcare professional are the key to an agreement on shared decisions about the patient's medicine. When healthcare professionals choose not to involve patients, they also deselect their patients' possibility to optimize their own health.

### Theme 3: Time pressure

Time constraints were a determinant that influenced patient involvement in decisions about medication in the ED. Time can be understood as both physical time (clock time) and mental time (the health professionals' experience of time) [34]. The results have shown that both clock time and mental time stressed the healthcare professionals, and lack of time was a constant focus both in the observations and during the interviews.

**Sub-theme: A revolving door.**   Several healthcare professionals explained that it was challenging to make time for medication conversations with patients in the ED due to the short admission periods and the fast patient flow, which often meant that patients were discharged or referred to other departments before decisions about the patients' medication were discussed with or even communicated to the patient. For example, a pharmaconomist described it this way:

> *"It's like a revolving door. Either the patient leaves the ED or is moved to another ward. So, the patient will probably be gone when you get back if you leave the ED to talk to a physician about the medication."* (Interview, Pharmaconomist, Department X).

**Sub-theme: Time consuming communication.**   Patient involvement includes more than patient information. A young physician stated during the interview:

> *"The physicians need to encourage the patient because the patients will not speak up unless we ask them."* (Interview, young physician, Department Y).

However, it takes time to establish a dialogue, and the healthcare professionals had the mental pressure that time was often not available because of the high patient flow. Therefore, asking patients questions, which is essential for patient involvement, was seen as a challenge. In the interviews, several healthcare professionals explained that they intentionally avoided asking the patients questions, as it could be time-consuming. For example, a nurse explained the dilemma: "*There is always pressure to see the next patient, so if they talk too much, we start wishing for them to finish because we know the next patient is already waiting.*" (Interview Nurse, Department X).

However, due to the healthcare professionals' mental focus on lack of time, it seemed this influenced the medication process and fragmented communication about medicine, which resulted in less effort to involve the patients.

### Theme 4: Faulty IT- systems

The observations revealed that the physicians spent noticeably more time on their computers than with their patients, which fragmented the medication process and communication about medicine and, hence patient involvement.

**Sub-theme: Poorly integrated IT systems lead to less patient involvement.** Poor integration between the SMC and the e-health platform made it difficult to obtain an overview of the patients' medication. In the interviews, several healthcare professionals explained that medication had become more complex and unmanageable after introducing the e-health platform. A pharmacist explained:

*". . . but it's obvious, that there are two IT- systems that don't work well together, and that they do not work well together does not make the situation concerning the patient's medication any easier . . . if just . . . if just all IT worked well, and it was possible to figure out what patients actually took at home. Often different medication is registered in the systems, and the larger the background issues, such as IT and such, . . . then focus is on those issues instead of the patient."* (Interview, Pharmacist, Department X).

This quote highlights how patient communication was often postponed due to IT issues in the office, and the observations revealed that medication decisions were often made by the physician in front of a computer before the patient was involved. This led to apparent fragmentation between medication decision-making and how patients were informed or involved in their medications.

The observations showed cultural differences in the two wards regarding how quickly physicians left the office to talk to patients. In Department Y, the chief physician acted as a role model and encouraged the other physicians to spend time with the patients rather than in front of the computer. The observations showed that the chief physician did so himself. In contrast, no chief physicians were observed encouraging patient interaction in Department X during the time of observation.

**Sub-theme: Challenging discharge.** The observations also showed that the challenges with IT- systems were most evident at discharge when the physicians, after reviewing the medications, should have re-entered the patient's medication from the e-health platform back onto their SMC. It was time-consuming, and physicians often expressed their frustration loudly. The physicians did not understand why they could not re-enter the medicine and the patient's usual medication in the SMC as prescribed and registered in the SMC by the GP before admission. This problem often made the physicians distinguish between medications concerning acute issues, which they were responsible for, and medicines prescribed by GPs, which they would not interfere.

However, one nurse explained in the interview that the problems were partly caused by themselves. They often forgot to decide the patient's medications registered in the SMC at the time of admission; thus, the problem was postponed to discharge. In addition, the nurse explained that this was not an isolated event:

*"Nobody makes a decision about medicine anymore. Many just go from SMC to the e-health platform."* (Interview, Nurse, Department X).

This shows that decisions about the patients' prior medications are not made at the time of admission and are therefore disconnected from the decisions made about medication in the ED, resulting in the patients risk of being discharged with a list of prescribed medications that have not been evaluated as a whole.

## Theme 5: The medicine list as a missed enabler of patient involvement

The observations showed that the patients' medicine lists as registered in the SMC or on the e-health platform were used inconsistently, but when they were used during ward rounds or during discharge, it created more dialogue, thereby enabling patient involvement.

Sub-theme: Inconsistently use of medicine lists

The physicians' only way to use the medicine list in conversations with the patient was by printing a copy of the medicine list. The interviews revealed that healthcare professionals valued the printed medicine list differentially. It was sometimes unclear who was responsible for printing and reviewing the medication list together with the patients. Several physicians stated that it was the nurses' or secretaries' job to print the medication list from the e-health platform and hand it to the patients at discharge. In contrast, several nurses believed this to be the physicians' responsibility.

Several physicians stated that they always printed the patient's SMC medicine list at admission and used it in conversations with the patients. However, the observations showed that the patients' printed medication lists were rarely present during ward rounds. The observations showed that in only a few cases did the physicians printed and used the e-health platform medication list at discharge. A nurse said:

*"I ask the physician to print it [medication list], but it often annoys them, and they do not have the printer login."* (Observation, Nurse, Department X).

In this, the observation revealed that the responsibility for printing the patient's medicine list was also fragmented and shared between the ED physicians and the nurses.

The interviews revealed that healthcare professionals had different explanations for why the medicine list was not used more frequently in medication conversations during ward rounds and discharge. One reason was that medication prescriptions had become electronic and paperless, so printing the patient's medicine list was obsolete. In addition, the observations showed that in contrast to the physicians, the pharmaconomists and nurses, when administering medications, frequently used a handheld rover (PDA) to show the patient a digital version of the medicine list to involve the patient visually.

In the interviews, several healthcare professionals elaborated that the lists were not used due to bad habits, a lack of attention, and a subculture in which healthcare professionals think others should involve the patient and review the medicine list with them. Asked why the medication lists were not used more often, an experienced chief physician answered:

*"I agree, but it is due to busyness and some very bad habits."* (Interview, Physician, Department X).

Nevertheless, the observations showed that the few times a printed medicine list was used to support medication conversations, the patient had several specific questions about the medications, resulting in more dialogue about medications, which is a prerequisite for patient involvement.

## Discussion

This study aimed to gain deeper insight into the determinants of patient involvement in decisions made in the ED about the patient's medication.

We found five themes (determinants) which affected patient involvement in decisions about their medication in the ED, which might hinder medication optimization. The five themes were: 1) blurred roles among multidisciplinary healthcare professionals, 2) older patients with polypharmacy increased complexity, 3) time pressure, 4) Faulty IT- systems, and 5) the medicine list as a missed enabler of patient involvement. In addition, four of the themes identified issues that were a barrier to patient involvement. In contrast, one of the themes, the

medicine list, identified a solution that could enable patient involvement if healthcare professionals were aware of this opportunity.

Other studies have also found that treating multimorbidity patients with polypharmacy is fragmented and influences patient involvement. Sinnott et al. [35] found that care provided by several medical specialists in the treatment of multimorbidity patients, compounded by poor coordination and communication within the health service and with the GP, was a challenge for patient-centered care and SDM. An Australian study [36] also found poorly defined individual responsibility and challenges with coordinating the treatment of older multimorbid patients with polypharmacy in community dwellings. As in our study, poorly defined responsibilities and challenges to coordination contributed to the avoidance of ownership for older patient's polypharmacy among healthcare professionals. The Australian study suggested delegating coordination and review responsibilities across healthcare sectors to care coordinators, which could be useful in improving overall care [36]. This highlights the fact that involving older patients with multimorbidity and polypharmacy in their medical treatment is not only a challenge in EDs but also for GPs and in communities.

One of our main findings was the perceived lack of time for patient involvement in medication in the ED. Légaré et al. [37] similarly found that time constraints, in general, are amongst the most frequently cited barrier to implementing SDM in clinical practice. However, one study found that the length of consultations varied from 8 minutes less to 23 minutes more (with a median of 2.5 minutes more) when the consultation was supported by a patient decision aid to facilitate SDM. This highlights the need for more research investigating whether patient involvement and SDM consumes significant amounts of time or less time [37] in the ED.

Probst et al. [14] also found that time is a core determinant of practicing SDM in the ED. The authors highlight that SDM should only be employed when time allows it, when the medical situation is nonurgent, and when there is no risk of poor patient outcomes due to the time taken to practice SDM. As our results revealed, the lack of physical (clock) time can make it more difficult for the healthcare professionals in the ED to find mental capacity to involve older polymedicated patients in decisions about their medicine if the patients are perceived challenging to involve, in the first place.

The medication process and medication conversations can be understood as "boundary objects" [30, 31]. According to the literature, a successful boundary object is able to transmit meaning between groups and provide a shared language to enhance knowledge [31]. The concept of "boundary objects" shares similarities with SDM and the aim of the medication conversation guide we plan to develop in a future study. The medication conversation guide aim to facilitate the sharing of information between patients and medical experts and the sharing of that information with other healthcare services, thereby contributing to the assurance of patient safety and patient involvement in medication, and hence medicine optimization. Our SDM medication communication guide, therefore, may have the potential to become a successful boundary object if it is well tailored to the ED context.

We did find one enabler for patient involvement in medication, which could also be characterized as a boundary object, e.g., the patients' printed medication list. When the printed medication list was actively used during medication conversations with the patient, we observed more dialogue between the patient and the healthcare professional than when the list was not used. We ascribe this dialogue to the use of the printed medication list. Unfortunately, the medication list was used sparingly and was often not updated by the patient's GP, which often caused even more confusion about the patient's medication even when the printed medication list was used.

Garfield et al. [38] also noted how the medication list has the ability to facilitate and change the way in which information about medicine was shared between patients and professionals. However, the study found significant barriers to the use of the medication list because patients did not understand the usefulness of the list. Professionals either lacked access to an accurate list or assumed that they had a good list, believing that different healthcare IT- systems communicated with each other, which—as in our findings—was not always the case.

Victoria Reay [39] highlights the so-called "productivity paradox", which had been a consequence of the digitalization of healthcare in the National Healthcare System (NHS) in the UK. It has led to workarounds, both within the computer systems and alongside them. Doctors are complaining about spending more time entering data than being with their patients [39]. Similar to results from our study, the physicians also spend a lot of time on faulty IT systems instead of having direct patient interactions indicating that if IT- systems in healthcare do not support clinical workarounds, it will become a barrier, rather than a facilitator for patient involvement.

We found that older multimorbidity patients with polypharmacy represent a substantial challenge in the ED. No one takes full responsibility for the patient's polypharmacy. The patients' medications were not accurately updated in the SMC, and the patients' medications became more confused as medication was added in the ED. This uncertainty about patients' medication made several ED physicians refrain from reviewing the patients' SMC medications and defer that review to the GP without involving the patient in this decision. This can be seen as a dilemma for ED physicians. According to the physicians' pledge, they have committed to "practice with conscience and dignity and following good medical practice" [40].

Patient-centered healthcare builds on the ideology that patients should become active partners in healthcare [41]. Furthermore, healthcare policies demand that healthcare professionals involve patients in their medical treatment [10, 42]. Therefore, it is essential for our society (and patients) that vulnerable patients with multimorbidity and polypharmacy be empowered to better care for themselves. However, at the same time, Ecks' recent study has shown that socioeconomic hardship and inequality make it more difficult for people from disadvantaged backgrounds to engage actively in treatment decision-making [43]. The importance of social inequality in multimorbidity is emphasized further in Barrett et al.'s study on multimorbidity, which reveals that the onset of multimorbidity occurs ten to fifteen years earlier in socially deprived areas, calling into question the existing single disease treatment approach in healthcare [44].

Nonetheless, a systematic review and meta-analysis indicate that SDM can improve outcomes considerably for disadvantaged patients. SDM has shown to enhance patients' knowledge and involvement in decision-making, reduce uncertainty about the course of action, and raise patients' decision self-efficacy [12]. As a result, it is argued that SDM has the potential to reduce health disparities significantly. Durand et al. [12] emphasize, however, the critical significance of tailoring design and content to the need of disadvantaged individuals if the potential of SDM is to be achieved.

Implementing patient involvement and SDM in everyday clinical practice is challenging if it is not applied to fit well into existing workflows and organizational systems [45]. Other SDM studies have found that culture is one of the most frequent barriers to SDM implementation [46]. According to Spradley, most ethnographers make use of what people say when trying to describe a culture. However, Spradley focuses more on making inferences from what people do (cultural behavior) and what they make and use (cultural artifacts), and the meaning they ascribe to these artifacts [23]. From our perspective, Spradley's methodology has been helpful for our study, as it has made it clear to us that artifacts—such as poorly integrated medication IT- systems and a lack of updated medicine lists—influence the culture surrounding patient

involvement in medication in the ED. As a result, our findings contribute to existing SDM research by looking beyond the direct interactions between patients and healthcare professionals and what people say about this and by highlighting the importance of exploring contextual and organizational aspects before introducing SDM in clinical practice, as these aspects may be strong determinants of patients' and healthcare professionals' ability to practice patient involvement in a specific context.

Furthermore, an updated review from the International Patient Decision Aid Standard collaboration (IPDAS) [47] investigated what works when implementing patient decision aids in routine clinical settings. The review found that patient decision aids are less likely to be used in typically crisis-driven teams or deal with life-threatening issues. The ED settings in our study often treat life-threatening conditions when the patient is admitted, but patients' conditions often stabilize within 24–48 hours after initial treatment, creating an opportunity for medication conversations with the patient. Additionally, it should be noted that 70% of all patients admitted to the EDs in our study was expected to be discharged directly from the ED, making this the best opportunity for medical conversations with the patient while hospitalized. Furthermore, our results revealed that the nurses and secretaries had essential but often invisible roles in the medication process even though the observations and interviews revealed that they often had much patient communication about medicine, and the nurses influenced the final decisions about the patient's medicine. Olling et al. [48] have in similar to our findings, found that oncology nurses play a crucial role in SDM as they support the patients' basic needs and help them navigate key decisions points as decision coaches, even though their work often is "invisible" [48]. If nurses and the health care team were more aware of–and thought to increase nurse's role in SDM, patient care would become more patient-centered [48]. This could also be relevant for the ED nurses and secretaries in the ED. Yet, it is relevant to explore if it is possible and reasonable to practice patient involvement and SDM with older patients with polypharmacy in the ED context and herby fulfil the healthcare policy goals of patient-centered healthcare.

## Methodological strength and limitations

This present study has several strengths and limitations. We have used the Standards for Reporting Qualitative Research (SRQR) [25], which we believe strengthens the validity and transparency of our ethnographic qualitative study. Our ethnographic approach is a strength. It enabled us to investigate how patient involvement in medication in the ED cannot be seen as an isolated phenomenon but is closely interrelated with context, healthcare technologies, and organizational pathways [23]. A large part of culture consists of tacit knowledge, which is most prominent in people's actions and not always visible in interviews. Therefore, combining field observations with semi-structured interviews with the healthcare professionals involved in medication processes in the ED is a strength [49]. Another strength is that the ethnographic fieldwork was performed by the same researcher, P.F., in two different ED settings with similar patient populations, which increases the generalizability of our findings, as it was possible to contrast the influence of local culture and influential key persons with more general findings [26]. This may have strengthened the transferability of our study results [50] to other ED settings.

There are also limitations. The purposeful sampling strategy [26] of the following healthcare professionals might have made the medication process appear more fragmented than it would have occurred had we chosen to instead follow a patient over time. On the other hand, our results provide insights into how fragmented and stressful managing medication can be for healthcare professionals in the ED. Furthermore, when performing ethnographic studies in a

hospital setting, the researcher can follow the healthcare professionals, the patients, or the visitors [51]. Most researchers choose to follow the staff (as we did)—a position that is well known to P.F., who has a professional background as a nurse. P.F.'s background may be a strength, but it may also be a limitation, as it may have shaped the focus on and the interpretations of specific themes, which another researcher may not have prioritized. To enhance the trustworthiness of our analysis, P.F. and J.K. frequently discussed the coding process and ethical issues to align interpretations of the data.

Furthermore, it might be a limitation that our ethnographic focus was mainly on the healthcare professionals and the context in which they work without focusing (much) on interactions with patients. However, patients are essential coproducers of patient involvement and influence how much (or little) involvement occurs, which may also have influenced our results. Therefore, the patient perspective will be explored in our following study.

## Conclusion

We found five themes (determinants) which affected patient involvement in decisions about their medication in the ED, which may hinder the patient's medication optimization. Four of the themes identified issues that prevented patient involvement. In contrast, one of the themes, the medicine list, identified a solution that could enable patient involvement if the healthcare professionals in the ED were aware of this opportunity.

A tailored medication conversation guide based on the SDM methodology may potentially function as a boundary object, supporting older polymedicated patients and healthcare professionals during medication reviews in the ED and across healthcare sectors to ensure that older patients receive medication aligned with their preferences and goals.

## Supporting information

**S1 File.**
(DOC)

## Acknowledgments

The authors would like to thank the management of both EDs for providing us with the opportunity to conduct field studies in their departments and to conduct interviews. We also thank all the healthcare professionals from both EDs for letting P.F. participate in their daily work.

## Author Contributions

**Data curation:** Pia Keinicke Fabricius.

**Formal analysis:** Pia Keinicke Fabricius, Jeanette Wassar Kirk.

**Funding acquisition:** Pia Keinicke Fabricius, Ove Andersen, Jeanette Wassar Kirk.

**Investigation:** Pia Keinicke Fabricius.

**Methodology:** Pia Keinicke Fabricius, Jeanette Wassar Kirk.

**Supervision:** Ove Andersen, Karina Dahl Steffensen, Jeanette Wassar Kirk.

**Validation:** Jeanette Wassar Kirk.

**Writing – original draft:** Pia Keinicke Fabricius, Ove Andersen, Karina Dahl Steffensen, Jeanette Wassar Kirk.

**Writing – review & editing:** Pia Keinicke Fabricius, Ove Andersen, Karina Dahl Steffensen,
Jeanette Wassar Kirk.

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
