## [Decision Letter · Decision Letter 0]

14 Jul 2021

PONE-D-21-12087

The challenge of involving old patients with polypharmacy in their medication during hospitalization in an emergency department: An ethnographic study

PLOS ONE

Dear Dr. Fabricius,

Thank you for submitting your manuscript to PLOS ONE. After careful consideration, we feel that it has merit but does not fully meet PLOS ONE’s publication criteria as it currently stands. Therefore, we invite you to submit a revised version of the manuscript that addresses the points raised during the review process.

The academic reviewers have provided a detailed review of the paper, and have raised a number of issues.  In particular: the naming of the themes and how they are presented in the paper, as well as strengthening the rationale for the chosen study setting, particularly regarding polypharmacy. There is also an inference in the paper that people with polypharmacy don't have access to, or undertake, SDM - I think this aspect of the paper could be strengthened in the background section.  Finally, please ensure sufficient context is given to ensure the significance of the work can be understood by an international audience.  

We look forward to receiving your revised manuscript.

Kind regards,

Adam Todd, PhD

Academic Editor

PLOS ONE

Journal Requirements:

Reviewers' comments:

Reviewer's Responses to Questions

**Comments to the Author**

1. Is the manuscript technically sound, and do the data support the conclusions?

Reviewer #1: Yes

Reviewer #2: Partly

2. Has the statistical analysis been performed appropriately and rigorously? 

Reviewer #1: N/A

Reviewer #2: N/A

3. Have the authors made all data underlying the findings in their manuscript fully available?

Reviewer #1: Yes

Reviewer #2: Yes

4. Is the manuscript presented in an intelligible fashion and written in standard English?

Reviewer #1: Yes

Reviewer #2: Yes

5. Review Comments to the Author

Reviewer #1: This is an interesting study of polypharmacy in two Danish emergency departments. It is based on a large set of interviews and field observations with different kinds of clinicians, nurses, and pharmacists. The key finding is that there are five dimensions that influence polypharmcy: blurred roles, old patients, time, IT systems, medicine as "missed enabler." The article needs substantial revisions before it can be published. First, the five dimensions are called "subthemes" but this seems an odd artifact of the qualitative data analysis. These aren't "subthemes" but rather important dimensions. What they are dimensions of should be clarified much more: that a patient gets more different medications prescribed, or that complicate treatment? These dimensions also need to be more clearly defined and specified, e.g. "time" can mean a lot of things. "Old patients with polypharmacy" wouldn't be a dimension, rather "relative age of patient" or something similar—"old patients" sounds a bit broad. The whole section on these dimensions, which are the key findings of the article, need a complete rewrite.

The methodology also needs works. In its current form, the article has an odd imbalance between far too much methodology described (down to examples of coding strategies, which I think goes beyond what is necessary for an article that tries to make an empirical contribution, not a contribution to methodology). Spradley gets a lot of prominence, which is odd, given that the works are from the 1970s and SO much has been written on fieldwork since. Why Spradley? On the other hand, the methodology sections do not explain why the authors chose two emergency medicine departments to study polypharmacy practices in the first place. Emergency departments are the least likely place to uncover "polypharmacy" — which is most important for long-term, chronic conditions and prescriptions. The multiple uses of medications for chronic conditions over long periods of time is precisely not what I would expect to find in an emergency department. So how did this study come about, did the authors set out to study emergency medicine and discovered polypharmacy as a problem along the way, or did they set out to study polypharmacy and then ended up in an emergency department for some reason? Indeed it's not clear at all what the original study design was, and even the section "study design" has nothing to say on this. Some of the places and institutions need to be made more specific, e.g. which "Institute of MEdicine"? It would be great also to learn more about possible Danish specificities in how healthcare is organized, maybe what goes under "emergency medicine" is slightly different than other countries. But there is nothing in the article yet that even explains the Danish healthcare infrastructure, and what place the studied locations hold within them.

Reviewer #2: This was a paper about shared decision making in the context of polypharmacy where patients were admitted to emergency departments of a hospital. Overall I think the methods (ethnographic) were conducted soundly but I wasn’t convinced about the naming of the main results finding ‘fragmented medicine’ – is it the medicine that is fragmented or the process in prescribing medicine? I found it hard to visualise what ‘fragmented medication’ was and how this could lead to fragmented patient communication (see abstract). Could it be that fragmented communication leads to fragmented medicine? Or maybe the authros when they are referring to communication they are only referring to communication in the ED?

But given that this in research uses an ethnographic and interpretive approach I would be happy to hear a bit more from the authors as to how they came to decisions to name the themes. I just found it hard to visualise the theme as it’s currently named. Additionally with the results I sometimes couldn’t follow through the main question/concern of the study which was about ‘patient involvement in medication in the ED’.

Here are some further comments about specific sections of the paper:

Abstract – background: I didn’t necessarily see the logical flow of argument between the sentences here. It’s not clear to me how patient involvement and SDM could be beneficial for patients with polypharmacy? Does it reduce polypharmacy? Does it help them manage high numbers of medicines?

Abstract – aim: Could the authors be more specific about what ‘patient involvement in medication’ means? Is this decisions about stopping/starting a medicine? Is it telling doctors about their medication and how it meets their goals? Is it about providing ED doctors with information about what medication they are prescribed and taking?

Introduction – I found this section the least well written of the paper and a bit muddled. I didn’t think it built up a persuasive argument about why the research was needed, what problem the ethnographic study would address. For example, the first sentence suggests that the majority of ED hospital admissions are acute and therefore the ED is a site for medication intervention, but I don’t see the logic in this. Are these 1,000,000 admissions medication related?

The second sentence is quite long with lot of things going on (definition of polypharmacy, statements about numbers of older people with polypharmacy/multimorbidity)

Another example where I think the authors could be more specific is page 6, line 87/88 ‘medicaiton should be more individualised’ - it’s not clear to me if the actual medicine or drug should be individualised (as in personalised medicine, pharmacogenetics’ or decisions about medication should be individualised (I imagine it’s the latter as the paper is concerned with shared decision making).

Materials and methods – perhaps a definition of SDM would be helpful (maybe it could go in the introduction (sorry if I missed it) and could be helpful to know what SDM means in the Denmark context and what it ideally looks like. This would then help the reader understand the statement on page 8, lines 16/117 ‘we did not expect to observe SDM in the ED, we focused on patient involvement…’ I think the authors might need to expand on the relationship between SDM and patient involvement.

Page 9, line 130 – define clinical pharmaconomists. I’ve never come across this term before! I looked it up is it like a UK pharmacy technician? Anyway would be good to know about it as seems quite a specific role that might be unique to Denmark?

I didn’t fully understand the setting - one hospital but two ED departments? This seemed unusual to me as my experience is that one hospital has one ED but of course it could be very different in Denmark. So some context would be helpful – what do ED departments do, how do they operate in the context of the hospital, why were there two EDs in this hospital

Participants – I just wondered if the authors sought out the participants in advance or came across them naturally as you were doing your observations.

Through the section field observations I started to get a better idea of what the researchers were looking at in terms of SDM/patient involvement so maybe some of this could also be in the introduction?

Results – as I said earlier I didn’t fully understand the overall theme ‘fragmented medication’. Could the authors provide a fuller explanation of the themes aside from the sub themes which fall under it?

Results subtheme 1 – this theme and the data descriptions within it resonated with my experience of conducting ethnographic research on polypharmacy in primary care in the UK. I found the data interpretations trustworthy and reliable. I did wonder if the first paragraph in this section is actually a description of the overall theme and therefore would be helpful upfront at the beginning of the results section.

My only issue with this section is the relevance of patient involvement. The section is very professional focused (and I accept the argument for this as presented in the discussion section) but the term patient involvement is used quite a lot in the section but without any explanation of what actually happened. For example page 18, line 283 ‘the secretaries had the task of involving the patients in their medication’ but then we don’t know what this involvement is.

There’s a lot in this section about no professionals wanting to accept responsibility for medication which I agree is the problem with trying to address polypharmacy, but what’s the link between professionals not accepting responsibility and patient involvement? The link needs clearly spelling out. OR was it the case that professionals thought it was the responsibility of other professionals and patient involvement was absent?

Results subtheme 2 – I found the contrast between older patients knowledge about the medicines versus the SMC fascinating. Neither was a particularly a source of trust for the ED doctors. But what I didn’t understand is by involving patients/SDM what were the ED staff hoping to achieve? Was there actually any involvement here or is the assumption that older people can’t be trusted mean they don’t actually get involved? So therefore is it assumptions about older people that are the problem not fragmented medication? Or that these assumptions lead to fragmented medication. I need a bit more convincing about what the relationship here is between this theme and the overall theme.

Results subtheme 3 – really liked this section and again resonated with work I’ve done with GPs and the problem of time pressures and conversations about medicines. But I did wonder how much can the ED be expected to have these discussions when it’s a revolving door? What kind of involvement can be expected? Might be useful to have something on what patient involvement in medication expectations in the ED are?

Discussion – I still found the relationship between the overall theme, patient involvement and the subthemes as determinants tricky to follow (para 1). I really liked the section on boundary objects and think this could contribute to understanding polypharmacy. Maybe hint of this theoretical lens at the outset of the paper could help orient the reader? Was the theory something that came to the data analysis early on/before data collection? The authors might be interested in the work of Victoria Reay about why we won’t have a paperless NHS anytime soon (https://www.lancaster.ac.uk/health-and-medicine/about-us/people/victoria-reay and see her article in the Conversation UK). Particularly in light of increased use of electronic records and electronic prescriptions.

6. PLOS authors have the option to publish the peer review history of their article (what does this mean?). If published, this will include your full peer review and any attached files.

Reviewer #1: No

Reviewer #2: **Yes: **Nina Fudge

---

## [Author Response · Author response to Decision Letter 0]

1 Oct 2021

The academic reviewers have provided a detailed review of the paper and have raised a number of issues. In particular: the naming of the themes and how they are presented in the paper, as well as strengthening the rationale for the chosen study setting, particularly regarding polypharmacy.

Thank you very much for letting us elaborate on the comments. We will try carefully to answer all questions in the following sections.

1. The naming of the themes and how they are presented in the paper,

 Answer:

Thank you very much for making us aware that we have been unclear in our first manuscript draft. In our revised manuscript, we have presented the former five sub-themes as themes (instead of one overall theme). We choose to call them themes as according to Braun and Clark thematic description of thematic analysis, we have followed: "a theme captures something important about the data in relation to the research question and represents some level of pattern or meaning within the dataset." 

The themes are presented in the following text added on page 15 : 

We identified five themes (determinants) which affected patient involvement in decisions about their medication in the ED:

1) Blurred roles among multidisciplinary healthcare professionals,

2) Older patients with polypharmacy increase complexity,

3) Time pressure,

4) Faulty IT- systems, and

5) The medicine list as a missed enabler of patient involvement.

See also our response to reviewer 1= A). and reviewer 2 =8).

2. The rationale for the chosen study setting, particularly regarding polypharmacy

Answer:

Thank you for placing this very relevant question. We have added the following text to reviewer 1 = 5):

"Older patients with polypharmacy are frequently admitted to medical EDs, with the need for a thorough medication review, and 85 % of the older polymedicated patients in our ED have potentially inappropriate medicine use. Because of the complexity of treating patients with polypharmacy in our ED, we have many ongoing studies investigating methods to improve the treatment of this large group of polymedicated patients. 

Furthermore, 70 % of all acute patients are discharged directly without further hospitalization in a medical ward in the future acute medical unit (ED). This also applies to the acute admitted older patients with multimorbidity and polypharmacy. 

Therefore, we find it relevant to carry out our study in our medical ED."

See also our answers to reviewer 1=5).

There is also an inference in the paper that people with polypharmacy don't have access to, or undertake, SDM - I think this aspect of the paper could be strengthened in the background section. 

Answer

Thank you for raising this point. Based on the comment, we have added the following text in the introduction section on page 5.

"The healthcare professionals' attitude towards patient involvement is also cited as a barrier because some healthcare professionals may have the assumptions that the older patients cannot contribute to decisions about their medicine, or the healthcare professionals think that they know their patient's preferences already so they do not need to ask the patient[16]. Additionally, older patients may have different preferences for involvement, low health literacy, difficulty hearing, or cognitive impairment, making it more challenging to become involved in decisions about medicines [16]".

 Finally, please ensure sufficient context is given to ensure the significance of the work can be understood by an international audience. 

Answer

Thank you for letting us elaborate on the danish ED context. We have also added the following text about the danish ED context to reviewer 1=7):

"7). We have now extended text about the Danish healthcare system with this text:

" The Danish welfare state provides free treatment for all citizens requiring medical care and free hospital and home-based care services. The acute medical patients are referred to the hospital either by the general practitioners (GPs), emergency medical helplines, after-hours GP services, or ambulances". 

The above text is added in the study setting section on page 6.

Furthermore, we have also added a detailed description of the ongoing reorganization of the Danish healthcare system and the Introduction of the new and more effective EDs, which is like healthcare policy reforms in other countries. 

Finally, we have added text that our ethnographic study explores new ways to optimize the acute patient pathways for older polymedicated patients before introducing the new EDs. 

"For over a decade, emergency services across Denmark have changed due to health policy reforms with the implementation of new Eds (so-called Acute Medical Units) to improve quality and efficiency of emergency care [21]. The new EDs offer a single point of hospital entry for all emergency care (except children and women in labor), 24/7 for effective emergency diagnostics and treatment with the continuous presence of senior physicians, covering ED physicians and senior physicians from the medical specialties. 

Several other countries as UK, Australia, the Netherlands, Germany, and France, have introduced similar reforms to secure safe and efficient pathways for patients in need of emergency care [22].

The new Danish EDs treat patients for up to 48 hours before discharge to home or a specialty department. However, it is the goal that 70 % of all acute patients are discharged directly from the new ED without further hospitalization to a medical ward. This also includes the medical treatment of older patients with multimorbidity and polypharmacy".

The above text is added on pages 7 and 8 the top section.

Answer:

Thank you for this important comment. 

We have added the following text in our cover letter at the very beginning.

 "The qualitative data of this study consist of danish field notes and interview transcripts which contain pseudo-anonymized potentially identifying information about the healthcare professionals, their managers, and the two included EDs. Even though the participants' names and identities are anonymized, it is still possible for local danish people from our Capital Region of Denmark to identify the ED departments included in the data. Unfortunately, the data are not publicly available due to ethical and legal restrictions on pseudo-anonymized data". 

Don't hesitate to get in touch with the research Data Protection Agency of the Capital Region of Denmark for further requests:

E-mail: videnscenterfordataanmeldelser.rigshospitalet@regionh.dk or Phone: +45 35 45 52

RESPONSE TO REVIEWER #1: 

Reviewer #1 Comments:

Reviewer 1: This is an interesting study of polypharmacy in two Danish emergency departments. It is based on a large set of interviews and field observations with different kinds of clinicians, nurses, and pharmacists. 

 The key finding is that there are five dimensions that influence polypharmcy: blurred roles, old patients, time, IT systems, medicine as "missed enabler." 

A). The article needs substantial revisions before it can be published. First, the five dimensions are called "subthemes" but this seems an odd artifact of the qualitative data analysis. These aren't "subthemes" but rather important dimensions. 

Answer:

A). Thank you very much for the comment. In our revised manuscript, we have presented the former five sub-themes as themes (instead of one overall theme). We choose to call them themes as according to Braun and Clark thematic description of thematic analysis, we have followed: "a theme captures something important about the data in relation to the research question and represents some level of pattern or meaning within the dataset." 

Furthermore, we realize that the naming of the sub-themes and the presentation have been unclear. Therefore, we have now rewritten most of the results section, which hopefully appears specific and clear. Our study aim was "To explore the determinants of patient involvement in decisions made in the ED about the patient's medication," which means that we have explored barriers and facilitators (determinants) for patient involvement in decisions about medicine.

From our perspective, all five themes either prevented or inhibited patients' involvement in decisions about their medication made in the ED. Herby, the themes can be ascribed as determinants for patient involvement.

See the result section, page 15-25.

B). What they are dimensions of should be clarified much more: that a patient gets more different medications prescribed, or that complicate treatment? These dimensions also need to be more clearly defined and specified, e.g., "time" can mean a lot of things. "Old patients with polypharmacy" wouldn't be a dimension, rather "relative age of patient" or something similar—"old patients" sounds a bit broad. The whole section on these dimensions, which are the key findings of the article, need a complete rewrite

Answer:

B). Thank you for raising this important question. In the revised manuscript, we have tried to describe how the determinants affected the patient's involvement in decisions about medical treatment in various ways. 

However, this study was an ethnographic study. Therefore we did not follow up on each patient's medical treatment, so we cannot describe the medical consequences of patient involvement or lack of involvement.

In our revised manuscript, we have now changed the names in themes 2-4 to: 

"Theme 2.: Older patients with polypharmacy increase complexity"

"Theme 3: Time pressure."

"Theme 4: Faulty IT-systems"

See result section page 15-25.

c). The methodology also needs works. In its current form, the article has an odd imbalance between far too

much methodology described (down to examples of coding strategies, which I think goes beyond what is necessary for an article that tries to make an empirical contribution, not a contribution to methodology). 

Answer:

c) Thank you for your comments. We have now removed 2 ½ lines on page 9. However, according to the standards for reporting qualitative research (SRQR) checklist, which recommends a detailed and transparent reporting of the data collection and analysis, we have tried to live up to these quality criteria for reporting on qualitative research.

However, we have tried to expand other sections in the manuscript to work with the balance. For example, we have extended the Introduction- and study setting section and Theme 1 in the result section.

See pages 3-5, 6-8, and 16-18.

4). Spradley gets a lot of prominence, which is odd, given that the works are from the 1970s and SO much has been written on fieldwork since. Why Spradley?

Answer:

4). Thank you for raising this question. We have chosen Spradley to guide our fieldwork primarily for two reasons. 

First, "Spradley is a classics within ethnographic fieldwork. Furthermore, Spradley offers a structured stepwise approach to participant observations, which is very helpful for ethnographers".

The above text has been added to page 10.

5). On the other hand, the methodology sections do not explain why the authors chose two emergency medicine departments to study polypharmacy practices in the first place. Emergency departments are the least likely place to uncover "polypharmacy" — which is most important for long-term, chronic conditions and prescriptions. The multiple uses of medications for chronic conditions over long periods of time is precisely not what I would expect to find in an emergency department.

So how did this study come about, did the authors set out to study emergency medicine and discovered polypharmacy as a problem along the way, or did they set out to study polypharmacy and then ended up in an emergency department for some reason? Indeed, it's not clear at all what the original study design was, and even the section "study design" has nothing to say on this. 

Answer:

5). Thank you for placing this very relevant question. 

Older patients with polypharmacy are frequently admitted to medical EDs, needing a thorough medication review, and 85 % of the older polymedicated patients in our ED have potentially inappropriate medicine use. Because of the complexity of treating patients with polypharmacy in our ED, we have many ongoing studies investigating methods to improve the treatment of this large group of polymedicated patients. 

Furthermore, 70 % of all acute patients are discharged directly without further hospitalization in a medical ward in the future acute medical unit (ED). This also applies to the acute admitted older patients with multimorbidity and polypharmacy. 

Therefore, we find it relevant to carry out our study in our medical ED.

We have added the following text in the introduction section on page 3.

"We have earlier identified that the use of potentially inappropriate use of medicine (so-called PIMS) in older patients (65 + years) admitted to our ED were very common (a prevalence of 85 %)".

 In the study setting section page 8 we have added the following text: 

"it is an overall goal that 70 % of all acute patients are discharged directly from the New ED without further hospitalization to a medical ward. This also includes the acute admitted older patients with multimorbidity and polypharmacy".

6.). Some of the places and institutions need to be made more specific, e.g. which "Institute of MEdicine"? 

Answer:

6). Thank you very much for making us aware of this. We have tried to be careful and more specific on places and institutions. For example, we have added a detailed description of the Declaration of Helsinki with the following: 

 "The World Medical Association Ethical principles for medical research involving human subjects. The Declaration of Helsinki".

The above text is added on page 14.

However, we have chosen to remove the text and reference on "the Institute of Medicine" in the introduction section because it referred to patient safety and IT systems in healthcare, which we have removed in this reversed manuscript due to a greater focus on patient safety in polypharmacy instead. 

7). It would be great also to learn more about possible Danish specificities in how healthcare is organized, maybe what goes under "emergency medicine" is slightly different than other countries. But there is nothing in the article yet that even explains the Danish healthcare infrastructure, and what place the studied locations hold within them.

Answer:

7). We agree that our description of the Danish healthcare systems was too superficial. 

We have now extended text about the Danish healthcare system with this text:

" The Danish welfare state provides free treatment for all citizens requiring medical care, as well as free hospital and home-based care services. The acute medical patients are referred to the hospital either by the general practitioners (GPs), emergency medical helplines, after-hours GP services, or ambulances". 

The above text is added in the study setting section on page 6.

Furthermore, we have added text to illustrate the high daily patient flow in the ED in the following text: "

"The EDs has an average daily intake of approximately 30-45 patients with hospitalization of up to 48 hours in the ED". 

The above text is added in the study setting section on page 7.

Furthermore, we have also added a detailed description of the ongoing reorganization of the Danish healthcare system and the Introduction of the new and more effective EDs, which is like healthcare policy reforms in other countries. 

Finally, we have added text that our ethnographic study explores new ways to optimize the acute patient pathways for older polymedicated patients before introducing the new EDs. 

"For over a decade, emergency services across Denmark have changed due to health policy reforms with the implementation of new Eds (so-called Acute Medical Units) to improve quality and efficiency of emergency care [21]. The new EDs offer a single point of hospital entry for all emergency care (except children and women in labor), 24/7 for effective emergency diagnostics and treatment with the continuous presence of senior physicians, covering ED physicians and senior physicians from the medical specialties. 

Several other countries as the UK, Australia, the Netherlands, Germany, and France, have introduced similar reforms to secure safe and efficient pathways for patients in need of emergency care [22].

The new Danish EDs treat patients for up to 48 hours before discharge to home or a specialty department. However, it is the goal that 70 % of all acute patients are discharged directly from the new ED without further hospitalization to a medical ward. This also includes the medical treatment of older patients with multimorbidity and polypharmacy".

"As the new ED is expected to be implemented in 2023, this baseline study serves as exploring new ways to improve the future patient pathways in the new EDs for older patients with polypharmacy, who are acutely admitted". 

The above text is added on page 7 and at the top of page 8.

RESPONSE TO REVIEWER #2: 

Reviewer #2 Comments:

Reviewer 2: This was a paper about shared decision-making in the context of polypharmacy, where patients were admitted to emergency departments of a hospital. Overall I think the methods (ethnographic) were conducted soundly but I wasn't convinced about the naming of the main results finding 'fragmented medicine' – is it the medicine that is fragmented or the process in prescribing medicine? I found it hard to visualise what 'fragmented medication' was and how this could lead to fragmented patient communication (see abstract). Could it be that fragmented communication leads to fragmented medicine? Or maybe the authros when they are referring to communication they are only referring to communication in the ED?

8). But given that this in research uses an ethnographic and interpretive approach I would be happy to hear a bit more from the authors as to how they came to decisions to name the themes. I just found it hard to visualise the theme as it's currently named. Additionally, with the results I sometimes couldn't follow through the main question/concern of the study which was about 'patient involvement in medication in the ED'.

Answer:

Thank you very much for raising this question. 

We know that we have not been clear enough about how we defined Fragmented Medication in our former version of the manuscript. We meant the Fragmented medication process, which in different ways influenced patient involvement in decisions about medicine in the ED. Therefore, we have changed the five sub-themes into themes, with minor changes in the names, but the content is almost the same. Included in the five themes, we have added text about the fragmented medication process and how it influenced patient involvement

"The five themes are:

1) Blurred roles among multidisciplinary healthcare professionals,

2) Older patients with polypharmacy increase complexity,

3) Time pressure,

4) Faulty IT- systems, and

5) The medicine list as a missed enabler of patient involvement."

The themes are presented with eight relating sub-themes in the result section from page 15-25.

9). Abstract – background: I didn't necessarily see the logical flow of argument between the sentences here. It's not clear to me how patient involvement and SDM could be beneficial for patients with polypharmacy? Does it reduce polypharmacy? Does it help them manage high numbers of medicines?

Answer:

9). Thank you for raising this question. We have not been precise in our description. We apologize.

We have now changed the text to the following text:

 "Research suggests that patient involvement and shared decision making (SDM) could optimize the treatment of older patients with polypharmacy as the patients become more aware of potential outcomes and therefore often tend to choose less medication". 

The text is inserted in the Abstract background section page 2.

10). Abstract – aim: Could the authors be more specific about what 'patient involvement in medication' means? Is this decisions about stopping/starting a medicine? Is it telling doctors about their medication and how it meets their goals? Is it about providing ED doctors with information about what medication they are prescribed and taking?

Answer:

10). We agree. It is patient involvement in decisions about medicine, and we have now changed the aim in the abstract (and in the manuscript) to the following text: 

"The aim was to explore the determinants of patient involvement in decisions made in the ED about the patient's medication.

The text is changed in the abstract and on pages 5, 25.

11) . Introduction – I found this section the least well written of the paper and a bit muddled. I didn't think it built up a persuasive argument about why the research was needed, what problem the ethnographic study would address. For example, the first sentence suggests that most ED hospital admissions are acute and, therefore, the ED is a site for medication intervention, but I don't see the logic in this. Are these 1,000,000 admissions medication-related?

Answer:

11). Thank you for making us aware that we have been too unclear in our description of the rationale for doing this study in an ED. We have now changed the Introduction section with more references and explanations about how many older patients with polypharmacy are acutely hospitalized in the ED and the complex treatment of this patient group. Furthermore, we have added text about the overall goal that 70 % of all acute patients in the new ED will be discharged directly from the ED without further treatment in a medical ward.

The following text is added in the introduction section:

"Globally, the prevalence of patients with polypharmacy is set to rise as the population ages, and more people suffer from multiple long-term conditions, which takes multiple medicines. This makes more people prone to drug-related events and frequently acute admissions in emergency departments. [1,2]. More than 1,000,000 out of 1,300,000 yearly hospital admissions in Denmark are acute admissions in emergency departments (EDs), and more than 70% of these admissions are older patients (65 + years) with comorbidity conditions and polypharmacy [3,4], defined as taking five or more medications [5]". We have earlier identified that the use of potentially inappropriate use of medicine (so-called PIMS) in older patients (65 + years) admitted to our ED was very common (a prevalence of 85 %). Furthermore, the use of PIMS was associated with low functional status and reduced health-related quality of life [7]".

The above text is added on page 3.

Furthermore, we have also added the following text to strengthen the rationale for conducting this study in the medical ED:

"The new Danish EDs treating patients for up to 48 hours before discharge to home or a specialty department. It is an overall goal that 70 % of all acute patients are discharged directly from the new ED without further hospitalization to a medical ward. This also includes the acute admitted older patients with multimorbidity and polypharmacy.

As the new ED is expected to be implemented in 2023, this baseline study serves as exploring new ways to improve the future patient pathways in the new EDs for older patients with polypharmacy."

The above text is added on page 8 in the top section.

12). The second sentence is quite long with a lot of things going on (definition of polypharmacy, statements about numbers of older people with polypharmacy/multimorbidity)

Answer:

12). We have changed the introduction section. See our reply in answer 11):

We hope this new introduction section is easier to read.

13). Another example where I think the authors could be more specific is page 6, line 87/88' medication should be more individualised' - it's not clear to me if the actual medicine or drug should be individualised (as in personalised medicine, pharmacogenetics' or decisions about medication should be individualised (I imagine it's the latter as the paper is concerned with shared decision making).

Answer:

13). We agree and have removed this reference and text from our manuscript as the reference was about personalized medicine and not about patient involvement or SDM. Again, we apologize that it created confusion. 

14). Materials and methods – perhaps a definition of SDM would be helpful (maybe it could go in the Introduction (sorry if I missed it) and could be helpful to know what SDM means in the Denmark context and what it ideally looks like. This would then help the reader understand the statement on page 8, lines 16/117 'we did not expect to observe SDM in the ED, we focused on patient involvement….' I think the authors might need to expand on the relationship between SDM and patient involvement.

Answer:

14). Thank you for this question. We completely agree and have added the following text and definitions in the introduction section: 

"The term "Patient involvement" is often used interchangeably and covers a diverse range of possibilities, emphases, models, and practices, with Shared decision Making (SDM) being one of the leading concepts [11]. SDM is cited to be the pinnacle of patient-centered care [12] and is a specific approach where healthcare professionals and patients build a partnership, where the healthcare professionals share the best available evidence, and the patients express their values and preferences and participate in decisions about their medical treatment [13].

Ideally, SDM with older polymedicated patients in the ED would be a process where the older patients are "active" partners, which implies that the patient is invited to ask questions about their medicine, and the patient receives information about the options they have, and decisions about changes (or no changes) to the patients' medicine are reached jointly [11]."

The above text is added on page 6 middle section.

15). Page 9, line 130 – define clinical pharmaconomists. I've never come across this term before! I looked it up. Is it like a UK pharmacy technician? Anyway, it would be good to know about it as it seems quite a specific role that might be unique to Denmark?

Answer:

15). Thank you for raising this question. We agree that we didn't introduce the role of pharmaconomist in our first draft of the manuscript. We have now updated the text with more details about Pharmaconomists in Denmark, which is comparable to the international term Pharmacy technicians.

See the inserted text: 

"Since 2014, Department X has had clinical pharmacy technicans (in Denmark called pharmaconomists) emploued who are responsible for the administration of medication during dayshift. The pharmaconomists hold a three-year degree and are comparable to most pharmacy technicians in other countries [23] In Denmark, pharmaconomists work at own private pharmacies and in-hospital pharmacies, where they often are responsible for the daily clinical management of medicine in the wards. Their work includes administering medicine to the hospitalized patients in the ward. Like the pharmacy technicians in the UK, the pharmaconomists in Denmark practice dispensing medicine without the supervision of a pharmacist".

The above text is added to page 8 at the middle section.

16). I didn't fully understand the setting - one hospital but two ED departments. This seemed unusual to me as my experience is that one hospital has one ED, but it could be very different in Denmark. So some context would be helpful – what do ED departments do, how do they operate in the context of the hospital, why were there two EDs in this hospital

Answer:

16). We have added a more detailed description of our hospitals and how the EDs are located.:

"The study was conducted at a university hospital in the capital region of Denmark where the healthcare system is publicly funded by taxes. The Danish welfare state provides free treatment for all citizens requiring medical care and free hospital and home-based care services. The acute medical patients are referred to the hospital either by the general practitioners (GPs), emergency medical helplines, after-hours GP services, or ambulances. 

The university hospital is divided into three different locations which each houses a medical emergency department to cover the catchment area of 517,00 people. The context in this study includes two of the medical emergency departments, which is named (Department X and Department Y).

See the setting section on page 6.

17). Participants – I just wondered if the authors sought out the participants in advance or came across them naturally as you were doing your observations.

Answer:

17). Thank you for raising this question. 

We have added the following text:

"The recruitment of participants was planned in advance with the departments managing nurses, who helped to locate participants who fulfilled our selection criteria of covering participants from different healthcare professions, gender and with most diverse years of clinical competencies. From the start, we had the assumption that all healthcare professionals in the ED (except the physiotherapists and secretaries) were involved in the medication process and therefore were relevant informants who were included based on their profession (ED physicians, geriatric physicians, consulting physicians from the medical specialties, nurses, pharmacists, and pharmaconomists). 

The above text is now added on page 11 in the top.

Furthermore, the following text has been added to the discussion section: 

"Our results revealed that the nurses and secretaries had important, but often invisible roles even though the observations and interviews revealed that they often had much patient communication about medicine and the nurses often influenced the final decisions about the patient's medicine. This point is added and discussed in the discussion section. Olling et al. have also found that oncology nurses play a crucial role in SDM. They support the patients' basic needs and help them navigate key decisions points as decision coaches, eventhough their work is often "invisible." If nurses and the health care team were more aware of – and thought to increase nurses' role in SDM, patient care would become more patient-centered. This could also be relevant for the ED nurses and secretaries in the ED".

See page 30 at the top section.

18). Through the section field observations, I started to get a better idea of what the researchers were looking at in terms of SDM/patient involvement, so maybe some of this could also be in the Introduction?

Answer:

18). Thank you for making us aware of this. We have now added the following text: 

"Ideally, SDM with older polymedicated patients in the ED would be a process, where the older patients are "active" partners, which implies that the patient is invited to work in partnership with the healthcare professionals about treatment choices. Different options are presented using risk communication, and the patient's preferences are explored and supported, and decisions about changes (or no changes) to the patient's medicine are reached jointly".

The above text is added in the introduction section on page 4 at the middle section.

However, at the beginning of the Materials and Methods section, we have added the following text:

"We did not expect to observe SDM in the ED as SDM has not yet, been implemented in the medical ED. We, therefore, focused on patient involvement, which is more broadly defined by Cribb [12] as "active rather than passive patients." Therefore, when exploring patient involvement in decisions about medicine in the ED, we focused on the patient's active role in decisions about their medicine in the medical ED and determinants influencing this active involvement".

The above text is added on page 6 in the top section.

19). Results – as I said earlier I didn't fully understand the overall theme 'fragmented medication'. Could the authors provide a fuller explanation of the themes aside from the sub themes which fall under it?

Answer:

19). We agree, and as we already have written, we have now removed the overall theme Fragmentated Medication and replaced the former five sub- themes as themes. In addition, we expanded with further text connecting each theme with patient involvement in decision about medicine it was possible.

See our reply in 8) and reply in 20-24).

20). Results subtheme 1 – this theme and the data descriptions within it resonated with my experience of conducting ethnographic research on polypharmacy in primary care in the UK. I found the data interpretations trustworthy and reliable. I did wonder if the first paragraph in this section, a description of the overall theme, is actually and therefore would be helpful upfront at the beginning of the results section.

Answer:

20). Thank you for making us aware of this. We have now added the following text in the beginning of the result section on page 15

"The observations and interviews revealed that managing medication to- and communication with the patients in the medical ED is a fragmented process that involves different healthcare professionals, different medical specialties, time pressure, and faulty IT- systems, which influence patient involvement in decisions about medicine and hinder medication optimization. Further, the study revealed that the patients printed medicine list could facilitate more communication about medicine and increase patient involvement in decisions about medicine in the ED if the healthcare professionals were more aware of this opportunity". 

21). My only issue with this section is the relevance of patient involvement. The section is very professional focused (and I accept the argument for this as presented in the discussion section) but the term patient involvement is used quite a lot in the section but without any explanation of what actually happened. For example page 18, line 283 'the secretaries had the task of involving the patients in their medication' but then we don't know what this involvement is.

Answer:

21). We agree and have now specified more what we meant with patient involvement. The following new text is an example of the added text. It is inserted on page 20 in the middle section: 

"The nurses had frequent conversations with the patients about medicine, and often it was the nurses who knew the reason why a patient did not take his or her medicine as intended. The nurses passed on the information to the physicians, who made the medication changes based on the information from the nurses. Therefore, nurses often had a decisively important influence in decisions about their patient's medication in the ED, even though many nurses were not always aware of it".

Furthermore, the following text has also been added:

"The interview revealed that the secretaries often had the task of follow-up communication about medicine with patients calling the ED after discharge who were not aware of why there had been changes to their medicine, which happened quite often according to the secretaries." 

The text above is added on page 17.

22). There's a lot in this section about no professionals wanting to accept responsibility for medication which I agree is the problem with trying to address polypharmacy, but what's the link between professionals not accepting responsibility and patient involvement? The link needs clearly spelling out. OR was it the case that professionals thought it was the responsibility of other professionals and patient involvement was absent?

Answer:

Thank you for this sharp question. 

We have added the following text:

22). 

" The physicians often preferred not to evaluate a polymedicated patient's usual medications as they did not always perceive that it was their job, but the patient's GPs job. In these situations, very limited patient communication about medicine was observed. Only when it was suspected that the patient's acute situation was caused by his or hers medicine, the ED physicians had many questions to the patient and information about the medicine".

The above text has been added to page 18.

23). Results subtheme 2 – I found the contrast between older patients' knowledge about the medicines versus the SMC fascinating. Neither was a particularly a source of trust for the ED doctors. But what I didn't understand is by involving patients/SDM what were the ED staff hoping to achieve? Was there actually any involvement here or is the assumption that older people can't be trusted mean they don't actually get involved? So therefore, is it assumptions about older people that are the problem not fragmented medication? Or that these assumptions lead to fragmented medication. I need a bit more convincing about what the relationship here is between this theme and the overall theme.

Answer:

23). Thank you for raising these very relevant questions. It was not all older polymedicated patients who weren't involved in decisions about their medicine, but the combination of high age and polypharmacy seemed to influence the healthcare professionals' attempt to involve the patients. It was both a matter of trust but also the complexity of the patient's polypharmacy that seemed to affect how little or much the healthcare professionals involved the patients. 

We have added the following new text on page 19 from one of the interviews:

"In the interview, a younger physician explained that he deliberately did not involve older patients with polypharmacy in decisions about their medicine because he had experienced that older patients often did not know about their medicine or did not care about it, and therefore he did not try."

And we have also added the following text:

 "However, some physicians did an great effort to involve the older patients with polypharmacy at the physicians round, and sometimes it seemed to be a success, and there was dialogue, and the patient had an influence on decisions about which medicine should be deprescribed before discharged. The observations revealed that older polymedicated patients who were asking questions about their medicine and seemed to have high health literacy were easier to involve because they often involved themselves. This highlights that older patients have different prerequisites for involvement. Relations between the patient and the healthcare professionals are the key to an agreement on" .shared decisions about the patient's medicine. When the healthcare professionals choose not to involve patients, they also deselect the patient's possibility to optimize their own healthcare.

The above text is added on page 20 at the top.

24). Results subtheme 3 – I really liked this section and again resonated with work I've done with GPs and the problem of time pressures and conversations about medicines. But I did wonder how much can the ED be expected to have these discussions when it's a revolving door? What kind of involvement can be expected? Might be useful to have something on what patient involvement in medication expectations in the ED are

Answer:

24). Thank you for raising this question. We have added the following text on page 8 at the bottom under the study setting section. We have added the following text, which illustrates that the pharmacists and geriatric doctors do structured medication reviews with the most complicated older polymedicated patients in the ED. So it is possible and relevant to explore the potential for implementing SDM and patient involvement in an ED.

"The Pharmacists and the geriatric physician do structured medication reviews on a daily basis for selected older polymedicated patients in the ED." 

Yet, it is relevant to exploreif it is possible and reasonable to practice patient involvement and SDM with older patients with polypharmacy in the ED context and herby fulfill the healthcare policy goals of patient-centered healthcare.

The above text is added to the discussion section on page 30 in the middle section.

25). Discussion –1.) I still found the relationship between the overall theme, patient involvement, and the subthemes as determinants tricky to follow (para 1). 2.) I really liked the section on boundary objects and think this could contribute to understanding polypharmacy. Maybe a hint of this theoretical lens at the outset of the paper could help orient the reader? Was the theory something that came to the data analysis early on/before data collection?

Answer:

25). We have removed the overall theme Fragmented Medication. Hopefully, it has made the result section easier to read and understand. Furthermore, we have now added an introduction to the theoretical lens of Boundary object in the analysis section on pages 13.

"During the analysis and interpretation process, it became clear to us that medication processes (and especially polypharmacy) cross boundaries between different healthcare sectors, different departments, medical specialists., and medication and can be understood and explained throughout the sociological theory of boundary objects. Boundary objects have the ability to transmit knowledge and meaning between different groups of people [34], and our results will be discussed based on this theoretical lens in the discussion section." 

See also the discussion section on page 27.

26). The authors might be interested in the work of Victoria Reay about why we won't have a paperless NHS anytime soon (https://www.lancaster.ac.uk/health-and-medicine/about-us/people/victoria-reay and see her article in The Conversation UK). Particularly in light of increased use of electronic records and electronic prescriptions.

Answer:

26). Thank you very much for the hint. Victoria Reay has very interesting work going on, which we definitely will follow in the future. We also have cited her work in our discussion section on page 28 in the top section.

"Victoria Reay highlights the so-called "productivity paradox," which had been a consequence of the digitalization of healthcare in the National Healthcare System (NHS) in the UK. It has led to workarounds, both within the computer systems and alongside them. Doctors are complaining about spending more time entering data than being with their patients [40]. In our study, the physicians also spend a lot of time on faulty IT systems instead of having direct patient interactions indicating that if It- systems in healthcare do not support clinical workarounds, it will become a barrier and not help to patient involvement".

Thank you very much for the detailed review.

---

## [Decision Letter · Decision Letter 1]

23 Nov 2021

PONE-D-21-12087R1The challenge of involving old patients with polypharmacy in their medication during hospitalization in a medical emergency department: An ethnographic studyPLOS ONE

Dear Dr. Fabricius,

Thank you for submitting your manuscript to PLOS ONE. After careful consideration, we feel that it has merit but does not fully meet PLOS ONE’s publication criteria as it currently stands. Therefore, we invite you to submit a revised version of the manuscript that addresses the points raised during the review process. Thank you for the time taken to address the comments raised by the reviewers.  Reviewer 1 makes an important role in acknowledging the role of the social determinants in shared decision-making, and how this is potentially more challenging for people of lower socioeconomic status.  I invite you to reflect on this comment, and perhaps acknowledge this in the discussion of your work.

We look forward to receiving your revised manuscript.

Kind regards,

Adam Todd, PhD

Academic Editor

PLOS ONE

Journal Requirements:

Reviewers' comments:

Reviewer's Responses to Questions

**Comments to the Author**

1. If the authors have adequately addressed your comments raised in a previous round of review and you feel that this manuscript is now acceptable for publication, you may indicate that here to bypass the “Comments to the Author” section, enter your conflict of interest statement in the “Confidential to Editor” section, and submit your "Accept" recommendation.

Reviewer #1: All comments have been addressed

Reviewer #2: All comments have been addressed

2. Is the manuscript technically sound, and do the data support the conclusions?

Reviewer #1: Yes

Reviewer #2: Yes

3. Has the statistical analysis been performed appropriately and rigorously? 

Reviewer #1: N/A

Reviewer #2: N/A

4. Have the authors made all data underlying the findings in their manuscript fully available?

Reviewer #1: Yes

Reviewer #2: Yes

5. Is the manuscript presented in an intelligible fashion and written in standard English?

Reviewer #1: Yes

Reviewer #2: Yes

6. Review Comments to the Author

Reviewer #1: This is a much improved second submission. The authors have worked very hard to address all the reviewers comments from the first submission, and the article is now almost ready to be accepted. I would recommend one more round of minor revisions to address a topic that is hugely important in the literature on multi-morbidity, which is social inequality or, more broadly, social determinants. The most cited study on multi-morbidity, Barrett et al. 2012 (doi: 10.1016/S0140-6736(12)60240-2) focuses heavily on the role of social determinants in relation to onset and severity of multi-morbidity, making the crucial observation that social deprivation can shift the onset of multi-morbidity foreword by 10 or 15 years. Social deprivation also has ramifications for shared decision-making, because patients from poor backgrounds find participation in decision-making so much harder, on so many levels. This has also been shown by several qualitative studies, e.g. recently by Ecks 2021 (https://link.springer.com/article/10.1007/s11013-020-09699-x). Since the focus of this article is on shared decision-making, at least a few lines should be devoted to the role of social determinants. It is quite possible that Denmark has less social inequality than other countries, and that therefore social determinants do not play a big role in this context, but even there I would have expected some discussion of it.

Reviewer #2: Thank you to the authors for addressing my comments so thoroughly. I recommend this paper for publication.

7. PLOS authors have the option to publish the peer review history of their article (what does this mean?). If published, this will include your full peer review and any attached files.

Reviewer #1: No

Reviewer #2: **Yes: **Nina Fudge

---

## [Author Response · Author response to Decision Letter 1]

1 Dec 2021

Dear 

Adam Todd, PhD

Academic Editor

PLOS ONE

Thank you very much for your and the reviewers' excellent response to our paper and for providing us with the opportunity to improve it. Please read the text box below for our responses to reviewer 1's comment.

Indeed, we do belive that socioeconomic disparity also is a rising concern in Denmark (as it is in other countries). We are fully aware of the challenges that multimorbidity patients from low-income families face when attempting to engage in treatment decisions. 

However, we do also believe that patients with inadequate health literacy still may have treatment preferences, eventhough they have difficulties in communicating with their physicians or pharmacist. Focusing on the patients preferences is an essential part of SDM and in this light, we hope to engage more multimoribidity patients in the medical treatment and health with SDM approach.

As requested, we also went over the reference list to confirm that none of the references are retracted. However, we have removed two references in our revised manuscript: 1) “Høj K, Mygind A, Livbjerg S, Bro F. Deprescribing of inappropriate medication in primary care. Ugeskrift for læger. 2019. And 2) Charles C, Gafni A, Whelan T. Decision-making in the physician-patient encounter: revisiting the shared treatment decision-making model. Soc Sci Med. 1999. We remowed the two references primarily because their content was similar to that of others references in the reference list, and to make room for two new references recommended by reviewer 1, which we have included in the disussion section to support the argument that people with lower socioeconomic baggrounds find it more difficult to participate in their treatment.

Response to reviewer 1:

Thank you very much for raising this important aspect and challenge in relation to SDM.

We agree with you, and have added the following text in the discussion section at page 28 bottom- section and page 29 top- section.

 It is essential for our society (and patients) that vulnerable patients with multimorbidity and polypharmacy be empowered to better care for themselves. 

“However, at the same time, Ecks' [2021] recent study has shown that socioeconomic hardship and inequality make it more difficult for people from disadvantaged backgrounds to engage actively in treatment decision-making [43]. The importance of social inequality in multimorbidity is emphasized further in Barrett et al.‘s study on multimorbidity, which reveals that the onset of multimorbidity occurs ten to fifteen years earlier in socially deprived areas, calling into question the existing single disease treatment approach in healthcare [44]”. 

We hope that you will find our revised manuscript suitable for publication in PLOS ONE.

---

## [Editor Report · Decision Letter 2]

6 Dec 2021

The challenge of involving old patients with polypharmacy in their medication during hospitalization in a medical emergency department: An ethnographic study

PONE-D-21-12087R2

Dear Dr. Fabricius,

We’re pleased to inform you that your manuscript has been judged scientifically suitable for publication and will be formally accepted for publication once it meets all outstanding technical requirements.

Kind regards,

Adam Todd, PhD

Academic Editor

PLOS ONE
---

## [Editor Report · Acceptance letter]

13 Dec 2021

PONE-D-21-12087R2 

The challenge of involving old patients with polypharmacy in their medication during hospitalization in a medical emergency department: An ethnographic study 

Dear Dr. Fabricius:

I'm pleased to inform you that your manuscript has been deemed suitable for publication in PLOS ONE. Congratulations! Your manuscript is now with our production department. 

Kind regards, 

on behalf of

Dr. Adam Todd 

Academic Editor

PLOS ONE